# Distortion Product Otoacoustic Emissions and Their Suppression as Predictors of Peripheral Auditory Damage in Migraine: A Case-Control Study

**DOI:** 10.3390/jcm10215007

**Published:** 2021-10-27

**Authors:** Maria Albanese, Stefano Di Girolamo, Lorenzo Silvani, Eleonora Ciaschi, Barbara Chiaramonte, Matteo Conti, Francesco Maria Passali, Battista Di Gioia, Nicola Biagio Mercuri, Arianna Di Stadio

**Affiliations:** 1Regional Referral Headache Center, Neurology Unit, University Hospital “Tor Vergata”, 00133 Rome, Italy; matteoconti92@gmail.com (M.C.); battista.di.gioia@uniroma2.it (B.D.G.); mercurin@med.uniroma2.it (N.B.M.); 2Department of Systems Medicine, University of Rome “Tor Vergata”, 00133 Rome, Italy; 3Unit of Otorhinolaryngology, Department of Clinical Sciences and Translation Medicine, Tor Vergata University, 00133 Rome, Italy; sdigirolamo66@gmail.com (S.D.G.); silvani.lor@gmail.com (L.S.); eleonora.ciaschi@gmail.com (E.C.); passali@med.uniroma2.it (F.M.P.); 4INAIL Consulenza Statistico Attuariale, Settore Tariffe, 00143 Rome, Italy; b.chiaramonte@inail.it; 5IRCCS Santa Lucia Foundation, 00179 Rome, Italy; 6Otolaryngology Department, Silvestri University Hospital, University of Perugia, 06151 Perugia, Italy; ariannadistadio@hotmail.com

**Keywords:** migraine, transient-evoked otoacoustic emissions, distortion-product otoacoustic emissions, cochlear damage, hearing impairment, olivocochlear system, CGRP

## Abstract

Although several cochleo-vestibular symptoms are commonly associated with migraine, only a limited number of studies have been done in this regard. Some reported abnormalities in audiometry, auditory brainstem response and vestibular tests, considering these manifestations mainly related to central etiology. However, increasing evidence also suggests a peripheral involvement of the inner ear in migraine. The aim of this study was to investigate the peripheral auditory pathway in migraineurs using otoacoustic emissions (OAEs), to detect alteration of cochlear functioning and possible relationship with disease severity. Sixty-two migraineurs and sixty matched controls were enrolled in the study and underwent a routine neuro-otolaryngology examination; self-administered questionnaires were used to evaluate subjective perception of hearing disability. DPOAE and their suppression were lower in migraineurs compared to controls and significantly related to the disease duration. Altered DPOAE exposed migraineurs to the risk of affecting by migraine without aura, of presenting with ocular and/or auditory symptoms during attack and of using more painkillers. Concomitant dopaminergic symptoms and/or allodynia such as the acute non-consumption of triptans were significant determinants of decreased contralateral suppression of DPOAE among migraineurs. This potential subclinical cochlear impairment in migraine detected by OAEs may represent the earliest sign of sensorineural damage in these patients, providing a promising tool for the initial diagnosis and an opportunity to monitor disease course and treatment response over time.

## 1. Introduction

Migraine is a debilitating condition characterized by unilateral throbbing head pain and a wide variety of other symptoms including photophobia, phonophobia, smell hypersensitivity, and a miscellany of autonomic, cognitive, emotional and motor disorders [1]. Many patients with migraine are also affected by cochleovestibular symptoms (i.e., fluctuating hearing loss, aural pressure, tinnitus, hyperacusis, vertigo, motion sensitivity and imbalance), with a prevalence of auditory problems [2]. Despite this finding, a causal link between migraine and auditory discomfort remains elusive and the authors have proposed several explanations for their coexistence. Vascular, neurochemical, genetic, cortical and central sensory processing have been speculated as causes of this observation [3]. Migraineurs present prolonged interpeak latencies in the auditory brainstem evoked responses (ABR) especially when they have documented hearing loss or vertigo, which might arise from peripheral or central origin [4]. Although patients with migraine complaining of hearing loss, pure tone audiometry and speech discrimination scores do not reveal any impairment in the most of cases [5]. Various abnormalities have been also reported in several vestibular testing such as electronystagmography, caloric examination, autorotation test, Romberg’s test, tandem Romberg test, and Fukuda stepping test; these alterations were present during the attack and in the interictal period, suggesting a possible peripheral inner ear involvement as cause of the labyrinthic dysfunction [6]. Several studies have investigated the pathophysiology behind the peripheral damage located in the inner ear in patients with migraine, using objective methods for hearing evaluation and assessment of the efferent system, i.e Otoacoustic Emissions (OAE), with conflicting results [5,6,7]. The olivocochlear (OC) system is considered the obliged pathway from the central nervous system to the cochlea and is divided in medial (MOC) and lateral (LOC) neurons [8]. MOC neurons are connected to the Outer Hairy Cells (OHCs) of the contralateral and ipsilateral ear, on the other hand LOC neurons are linked to ipsilateral dendrites of auditory nerve located below the inner hairy cells (IHCs) [8]. MOC and LOC cooperate in order to localize sound stimuli in noisy environment and integrate them. The OC system inhibits the activity of the OHCs, in fact, it can reduce the activity of basilar membrane, of the auditory nerve, and the intensity of otoemissions [8,9]. Collet et al. showed that studying the contralateral suppression of the OAE is the only non-invasive method to evaluate the MOC integrity [9]. Recently, growing evidence supports the hypothesis that migraine and inner ear pathology should not be merely considered as isolated entities (i.e., vestibular migraine or cochlear migraine), but they may exist on a continuum, resulting from dysfunction of interactions at many levels, from the cochlea to the brainstem throughout OC pathways, and thus to thalamus and cortical level [10].

Because OAE activity is related to the one of OC, we aimed to investigate the peripheral auditory pathway in untreated migraineurs by using OAEs, to detect alterations of cochlear functioning and the possible relationship with disease severity.

## 2. Materials and Methods

This observational case-control study enrolled sixty-two consecutive outpatients referred to the Headache Center at Tor Vergata University Hospital of Rome, who suffered from episodic migraine according to the International Classification of Headache Disorders 3rd edition (ICHD-3) [11]. The study was conducting in respect of Helsinki rules and all subjects signed a written consent before being included in the study. The Internal Committee of the University Hospital Tor Vergata approved the study with protocol number 28/2021. The study database is available from the Corresponding Author on reasonable request.

Inclusion criteria were: age between 18 and 50 years of age, not under treatment with prophylactic drugs against migraine, free of migraine attack (and symptoms-related) at least 3 days before examination. Patients of both gender who filled all the eligibility criteria were selected and included in the study.

Exclusion criteria were: treatment with corticosteroids up to 30 days before the neuro-otological examination, other neurological and psychiatric disorders, other primary form of headache, use of central nervous system-active medications or contraceptives, previous history of otological/labyrinthic disorders, exposure to excessive noise, ototoxic drug consumption, history of head or ear trauma, any systemic, cardiovascular, metabolic or autoimmune disorder which could be associated with hearing loss (i.e., renal insufficiency, gout, diabetes mellitus, hypertension, ischemic heart disease).

60 healthy volunteers, matched for age and sex, were recruited from hospital staff, friends, and colleagues as control group. Potential controls were excluded if they had a history of primary or secondary headaches, a history of otologic disease or clinically significant audio-vestibular symptoms, or any active medical disorder likely to interfere with testing.

All 122 people (patients and controls) were evaluated at the time of the enrollment by a neurologist and an otolaryngologist.

*Neurological examination*: Patients were assessed by a headache expert neurologist with a face-to-face interview using a semi-structured questionnaire addressing socio-demographic factors, clinical migraine features, previous and current acute and preventive migraine treatments, comorbidities, and concomitant medications. Enrolled patients were requested to report monthly migraine days (MMDs) and monthly painkillers intake (MPI). Disease duration was estimated as the number of years from onset to the most recent assessment. Headache-related dopaminergic and autonomic symptoms, temporal artery turgidity/hyperpulsatility, ocular and auditory manifestations, the occurrence of aura and allodynia during or between attacks were investigated.

*Otolaryngology investigations*: patients and control were analyzed by otoscopy, pure tone audiometry (PTA), acoustic immittance test, otoacoustic emissions (OAEs): transient-evoked (TEOAE) and distortion-product (DPOAE). Furthermore, an extensive otological anamnesis was performed to exclude any type of previous disease, i.e., external otitis with discharge.

Additionally, migraineurs and controls were asked to complete a self-report questionnaire designed to provide information about any hearing problems in normal everyday life situations [12,13] for investigating subjective perception of hearing disability.

### 2.1. Details of Audiological Investigations

An operator blinded to diagnostic category (i.e., migraine versus controls) performed the following auditory tests.

*Audiometry*: All tests were performed by the same audiologist in a silent booth. Healthy subjects underwent Pure-tone audiometry (PTA) testing to exclude hearing disorders. PTA was assessed for all subjects and measured with a calibrated dual channel mr27A EARTECH resonance audiometer at frequencies of 125, 250, 500, 1000, 2000, 4000, and 8000 Hz in a soundproof booth following the modified Hughson-Westlake ascending method [14]. Hearing loss was calculated separately for each pure tone frequency stimulation as well as the amount of threshold shift above the standard audiometric zero (dB HL).

*Impedancemetry*: A complete acoustic immittance test with tympanogram and acoustic reflex was performed in order to exclude possible middle ear illness (e.g., otosclerosis, glue or tympanic perforation) (GN Otometrics Zodiac). The intensity threshold of the acoustic reflex was determined for each ear using 500 Hz, 1000 Hz, 2000 Hz and 4000 Hz stimulus tones. The stimulus was presented either to the same ear as the compliance probe (ipsilateral reflex) or to the opposite ear (contralateral reflex).

*Otoacoustic emissions*: In the same session, Transient-evoked otoacoustic emissions (TEOAE) and distortion product otoacoustic emissions (DPOAE) were also recorded in both ears of all subjects and analyzed, using an ILO V6 Otodynamics Software (MAICO MI 34; Berlin, Germany) analyzer [15]. All measurements were performed in a quiet environment. Patients were asked to remain still and breath normally. Plastic tube adapters were fitted over probes that housed sound sources and microphones to ensure proper fitting of the probes in the external auditory canals.

TEOAE and DPOAE were investigated using a standard nonlinear click stimulus of 80-µsec duration that was presented at a repetition rate of 50 Hz and a sound level 80 (±3) peak equivalent dB SPL. The level of random noise was regulated by setting the automatic noise rejection at 47.3 dB in all tested subjects. Frequencies of 1000, 1500, 2000, 3000, 4000, 6000, and 8000 Hz were recorded. The stimulus consisted in two pure tones (F1, F2; F1/F2 = 1.22) at a sound pressure level of 70 dB. The signal-to noise ratio was reported.

TEOAE and DPOAE suppression was also tested. For TEOAE and DPOAE suppression testing, white noise was generated by a dual channel mr27A EARTECH resonance audiometer and was presented to the contralateral ear through a TDH 39 headphone. The frequency of the white noise was fixed at 1000 Hz with and intensity of 60-dB HL.

TEOAE and DPOAE suppression was measured by the probe in the right ear, while noise was delivered through the earphone placed in the left one. After switching the device sides, the procedure was repeated. The difference in the TEOAE amplitude with and without contralateral noise was referred to as total suppression (TS).

### 2.2. Statistical Analysis

Calculations were performed using the Statistical Package for the Social Sciences Windows, version 25.0 (SPSS, Chicago, IL, USA). Independent two-sided Student t test was used for comparison of the means of normally distributed measures. As a priori analysis, non-parametric tests, and contingency table (Chi-square and two-tailed Fisher exact tests) and unadjusted odds ratios (OR) with their 95% confidence intervals (CI) were run to compare variables between migraine patients and controls. Univariate and multivariable analyses using logistic regression models were performed to determine contributing factors for OAE in migraine patients. All tests were bilateral. Statistical significance was set as two-tailed *p* < 0.05.

## 3. Results

Twelve migraine patients and two controls were not included in our data analysis/dropped out: due to evidence of middle ear diseases during the otoscopic and/or acoustic impedance tests.

The final sample consisted of fifty migraineurs (mean age 36 ± 10 years; 15 males and 35 females) and fifty-eight control subjects (mean age 35 ± 8 years; 20 males and 38 females). The Migraine group, by history, showed a mean disease duration of 17 ± 12 years (range, 3–40 years) and, on clinical examination, 15 had migraine with aura (MA) and 35 had migraine without aura (MoA). All the MA patients had only visual aura (not accompanied by any other transient neurological deficits).

Ear fullness was the auditory symptoms encountered in patients with migraine during attacks, with estimated frequency of 22% (11/50 pts). No differences in gender or age were detected between the migraine patients and controls.

The demographic and clinical characteristics of the studied groups are shown in Table 1.

All subjects completed audiological evaluations and data were recorded from both ears. Otoscopic examination and acoustic impedance test revealed that all subjects had intact ear drums, a type “A” (normal) tympanogram with normal middle ear pressures and normal static compliance. Acoustic reflex was present bilaterally at 1 and 2 kHz at presentation levels not exceeding 100 dB HL.

No statistically significant differences were identified in audiometric measurements between the two ears in either group (migraine patients and control group).

The hearing disability questionnaire did not reveal a subjective perception of hearing impairment in both patients and controls group during the attack-free periods.

TEOAE and TEOAE suppression scores did not show statically significant difference between patients and controls.

On the contrary, DPOAE and contralateral suppression of DPOAE responses in all the frequencies studied tended to be lower in migraine patients compared to controls, with statistically significant value (see Table 2, Figure 1 and Figure 2). Compared with healthy control subjects, patients with migraine reported significant lowering mean DPOAE amplitudes observed at each frequency (*p* = 0.01 left, *p* = 0.00003 right). Contralateral sound stimulus induced significant decrease in amplitudes of DPOAE (suppression) in migraineurs compared to controls (*p* = 0.004 left, *p* = 0.00002 right) (see Table 2).

To better investigate the correlation between audiological variables and demographic and clinical characteristics, migraineurs were categorized according to the absence or the presence during the attack of: (1) aura; (2) headache-related dopaminergic and autonomic symptoms; (3) temporal artery turgidity/hyperpulsatility; (4) ocular and/or auditory manifestations; (5) allodynia.

The monthly use (average number) of painkillers for acute treatment of the headache episode, including triptans, was also considered.

The regression analysis showed that the reduction of the scores of both DPOAE and their contralateral suppression were significatively related to the length of disease affection (OR 0.020 and OR 0.076, respectively; 95% CI 0.001–0.580 and 95% CI 0.007–0.771, respectively; *p* = 0.023 and *p* = 0.029, respectively). Altered DPOAE (and suppression) total responses exposed migraineurs to the risk of presenting with ocular (i.e., ocular ptosis) (OR: 357.553; 95% CI 2.114–60,461.505; *p* = 0.025) and/or auditory symptoms (OR: 1100.476; 95% CI 2.612–4.636; *p* = 0.023) (i.e., fullness) during the headache attack (see Table 3). In case of reduced DPOAE the patients had an increased risk of suffering from migraine without aura (OR 0.025; 95% CI 0.001–0.466; *p* = 0.013). Furthermore, the lowest scores of DPOAE exposed the subjects to a highest consumption of painkillers (monthly average number) (OR 0.007; 95% CI 0.000–0.670; and *p* = 0.033) (see Table 3).

Multivariable analysis showed that the presence of dopaminergic manifestations (OR, 10.960; 95% CI 0.980–122.529; *p* = 0.052) and/or allodynia (OR, 25.136; 95% CI 1.710–369.467; *p* = 0.019) during attack such as the acute non-consumption of triptans (OR, 0.024; 95% CI 0.002–0.281 *p* = 0.003) were significant determinants of decreased contralateral suppression of DPOAE responses among migraineurs (see Table 3).

Furthermore, otoacoustic emission results were similar among patients categorized according to the side of pain (unilateral: *n* = 30; bilateral: *n* = 20) (not shown).

## 4. Discussion

Our study identified a difference in the value of DPOAE and DPOAE suppression between patients with migraine and control group (healthy). The patients presented lower DPOAE scores compared to healthy controls; this alteration did not correspond to subjective perception of discomfort or alteration of PTA. The reduction of DPOAE values increased the risk of presenting migraine without aura but with ocular and auditory symptoms. Moreover, the patients who suffered from migraine by several years presented lowest DPOAE scores. Overall, these results highlight that DPOAE might be used to identify potential reversible hearing impairment in migraineurs; the presence of these alterations suggests that subtle sensorial affection may be an early sign of headache, a sort of additional *“aura”* detectable by electrophysiologic study only. The causes of these electrophysiological disturbances might be varying (Figure 3); however, both DPOAE and their suppression test seem to be sufficiently sensitive to detect them. The differences among migraineurs and healthy were also present during the interictal period, independent of the side of pain; this finding may explain that such modifications in the inner ear are stable, independently from the headache phase (active or not). The absence of the auditory deficit both self-reported and as results of the otolaryngology tests might indicate a subclinical cochlear dysfunction.

Although cochleovestibular symptoms, such as dizziness, imbalance, phonophobia, tinnitus, nystagmus, occur commonly in patients with migraine during and between attacks, only a limited number of studies have been done in this regard, exploring the relationship between migraine and auditory function [1,2,3,4]. Some authors reported abnormalities in audiometry, auditory brainstem response and vestibular testing, suggesting a preferential dysfunction of the afferent auditory system and of the central sensory processing mechanisms as the basis of hearing damage in migraine [3,4,5,16,17,18].

Recently, increasing evidence suggests that the peripheral nervous system may also contribute to the clinical and subclinical manifestation of migraine, so the efferent auditory olivocochlear network could be involved at this level [18,19]. The olivocochlear (OC) system is considered the obliged neuronal pathway from the primary auditory cortex to the brainstem and, in turn, to the cochlea and is divided into medial (MOC) and lateral (LOC) bundles [8]. The outer hair cells of cochlea (OHCs) mainly receive contralateral fibers from the MOC, which is directly related to the inhibition of otoacoustic emissions (OAEs), whose assessment permits to objectively monitor dynamic changes in cochlear responsiveness before functional and significant hearing loss occurs from any cause [8,9]. Data evaluating OAE levels in migraineurs are not consistent across the studies, most probably due to the small size and high variability of patient cohorts [4,5,6,7].

Hamed and colleagues [4] reported significant lowering of TEOAE and DPOAE amplitudes in migraineurs compared with healthy controls, and these abnormalities appeared to be more evident in patients affected by migraine with aura. In contrast, Joffily et al. showed no difference of TEOAE and DPOAE between patients with migraine and controls; however, these alterations were observed only in women with phonophobia and at the selected middle-range frequencies [7].

In our larger case-control study, we firstly identified a difference only in the DPOAE and not in the TEAOE; these scores were significantly lower in the migraineurs compared to controls independently from gender. This result suggests that DPOAE could be used to potentially screen and identify patients with migraine.

Different hypothesis might explain this finding. As first, if we consider neuroinflammation as potential cause of migraine [20,21], the liquid exchange among cerebrospinal fluid (CSF), which contains inflammatory compounds and perilymph, might drive these agents into the inner ear. The pro-inflammatory cytokines, which are toxic for the hair cells [22], could induce an alteration in the DPOAE. Karatas et al. proposed a link between a noxious intrinsic brain event and activation of the trigeminal pain fibers, involving the opening of Panx1 megachannels on stressed neurons, subsequent activation of the inflammatory pathways, and transduction of this signal to the trigeminal nerves around pial vessels [23]. This inflammatory event may be transferred from the brain into the inner ear; in fact, CSF and perilymph are strictly connected so the inflammatory cytokines and other proinflammatory molecules might arrive in the inner ear and induce the hair cells damage, which is reflected by the altered DPOAE (Figure 3). Because the inner ear has an area smaller than brain, lower concentration of cytokines is necessary to determine a damage; so, the answers of DPOAE might indicate an early pre-symptomatic and clinical migraine. This hypothesis is supported by the evidence that patients affected by migraine for longer showed lowest DPOAE scores. In fact, the level of cytokines into the inner ear might increase by the time of affection and be responsible of the worsening of DPOAE. Since DPOAE amplitudes worsen with age similarly to hearing thresholds, we can also speculate that migraine may aggravate age-related hearing loss and increase hearing impairment caused by daily exposure to noise over time, according to our findings that hearing dysfunction correlated with disease duration [24]. The inflammatory theory might also explain the peripheric symptomatology as ptosis and fullness that were observed in migraineurs with altered DPOAE and DPOAE suppression. Finally, these alterations might also be sign of neural hyperexcitability found in the brain of migraineurs, which affects both cortical and subcortical/peripheral auditory pathways [25].

In contrast with previous studies, we showed that patients affected by migraine without aura had a higher risk of developing DPOAE reduction. Existing preclinical data suggest that in migraine aura there are centrifugally propagating signals from cerebral cortex to surrounding tissues, possibly via CSF route, that in turn can extend to affect the cochlea as well, contributing to the neurogenic inflammation and the pain phase [25,26]. Intriguingly, in migraine without aura it has been postulated a primary site of extra-axial inflammation at peripheral level [25,26], that may play more of a long-term modulatory role by promoting firstly the peripheral sensitization of trigeminovascular afferents in the inner ear and their downstream thalamic targets on neurons receiving converging inputs from the cochlea, creating a positive feedback loop.

Nevertheless, our patients with migraine exhibited deficit in DPOAE suppression compared to healthy subjects, which points to migraine as a disorder affecting in general the peripheral structures and the medial OC efferent system’s integrity. Previous studies already analyzed the OAE suppression in migraineurs, but not using DPOAE-notable more sensitive to detect alteration especially in neuroinflammatory diseases [27]; moreover the previous studies have been conducted on small-sized sample. Although the total suppression amplitude was lower in the individuals with migraine, these alterations were significantly observed only in a subset of migraineurs, such as women with phonophobia at the selected frequencies of 1–1.5 kHz [7] or patients with vestibular migraine [28]. We think that our broader cohort might have contributed to the lack of difference in the TEOAE suppression among groups. Furthermore, there are some relevant differences between TEOAE and DPOAE and, consequently, between their suppression magnitudes [8,9]. In fact, TEOAE originate from the activation of whole cochlea and provide information limited to middle-frequency ranges 1–4 kHz. Compared to TEOAE, DPOAE allow the assessment of cochlear status in a frequency-specific manner and are very sensitive to measure larger and higher frequency ranges [8,9]. DPOAE may exist even in the case of severe hearing impairment (of up to 50 dB) and long-lasting over the time being still detectable two hours after death [8]. Because the two investigations are complementary, we used both TEOAE and DPOAE suppressions, for a detailed cochlear evaluation in migraineurs. Pathophysiological mechanisms underlying this hearing injury are still incompletely delineated. Considering migraine as disease supported by vascular mechanisms, the cochlea dysfunction could be due to temporary reduction of blood flow during vasospasm. It has been shown that also a temporary inner ear hypoxia can cause the sufferance of the hair cells with auditory deficits [29]. It is well known that OAE suppression has a protective role because inhibition of OHC contraction reduces the odds of cochlear damage and is indicative of efferent pathways integrity from the pons to the cochlea via MOC [8,9]. Interestingly, acetylcholine (Ach) plays a pivotal role for the adequate functioning of MOC. In the inner ear, Calcitonin gene-related peptide (CGRP) acts as an inhibitory modulator of Ach, by facilitating the phosphorylation of its receptor [30,31]. On the other hand, CGRP is crucial for the trigemino-vascular system, which is deeply involved in the pathophysiology of migraine [25]. We propose that the deficient suppression of auditory pathway detected in our sample of migraineurs may be due to increased CGRP activity in the inner ear, that acts as an inhibitory modulator of Ach at the cochlear synapses, especially in subjects with a longer disease duration. This phenomenon might account for the higher risk of significant reduction of contralateral DPOAE suppression between patients that did not use triptans during the attacks. Triptans, a specific symptomatic migraine treatment, are serotonin agonists and have greatest affinity for 5HT receptors that appear to be co-expressed with CGRP in vestibular and spiral ganglion cells [25,31]. Importantly, CGRP levels were reduced by triptans, coincident with pain relief, at both central and peripheral afferent terminals [25]. We speculate that the absence of triptans use for treating migraine attacks might facilitate CGRP release and its exitoxic and pro-inflammatory activities in the inner ear, resulting in imbalance between excitation and inhibition and leading to reduced activation of the MOC in the brainstem with further recurrent hearing damage [32]. In fact, as well as the enhancement of the MOC prevents hearing loss, the reduction of MOC activation could worsening the hearing capacity [32]. Moreover, because pain induces release of the pro-inflammatory cytokines [20,21], not using triptans might increase the concentration of these elements into the CSF; the physiologic liquid exchange between brain and inner ear (CSF and perilymph) carries the inflammatory cytokines into the inner ear causing damage [22]. (Figure 3).

Additionally, in our sample the OAE abnormalities were bilateral and independent by the side of pain, suggesting an activation of the peripheral patterns, mainly involving the brainstem and the neuromuscular structures [32]. This hypothesis is also confirmed by normal pure tone audiometric thresholds, so more subtle auditory changes in migraine may exist early in the disease course and OAEs, by examining the efferent system, are sensitive enough to identify them. Finally, the occurrence of allodynia and/or dopaminergic manifestations during attack, such as the high intake of painkillers, that have been identified as a potentially prognostic factor for more severe and refractory forms of headache or transformed migraine [25], increased the risk of DPOAE abnormalities among migraineurs. Thus, disruption in the contralateral suppression/OAE and OC malfunctioning may be one of the mechanisms predisposing to other clinical symptoms associated with migraine and contributing to its chronicization.

### Limitations of the Study

This study presents some limitations. The first is the sample dimension; we analyzed only 50 patients in the study group and other authors presented results on wide sample; however, we selected a population with episodic headache who were not under pharmacologic prophylactic treatment. As additional, the other authors reported information about TEOAE, while we described DPOAE and their suppression; it is well known the more relevant clinical meaning of DPOAE compared to TEOAE specially to diagnose subclinical pathology [27].

Another limitation was the non-inclusion of newly started migraine patients and triptan treated migraineurs; in the latter the benefit of the treatment could have an effect on the DPOAE results.

## 5. Conclusions

As previously demonstrated for other neurological diseases [27,33], our results confirm a potential cochlear impairment in migraineurs, occurring at synaptic level between MOC and OHCs and targeting CGRP activity. A subclinical cochlear discomfort and a concomitant detectable dysfunction of the OHCs by OAEs may represent the earliest sign of sensorineural damage in these patients, providing a powerful tool for the initial diagnosis and an opportunity to monitor disease course and treatment response over time. This is also particularly intriguing in relation to the newly emerging therapeutical options for migraine, such as CGRP antagonism/anti-CGRP monoclonal antibodies, that seem to have a net impact on disease disability and progression.

## Figures and Tables

**Figure 1 jcm-10-05007-f001:**
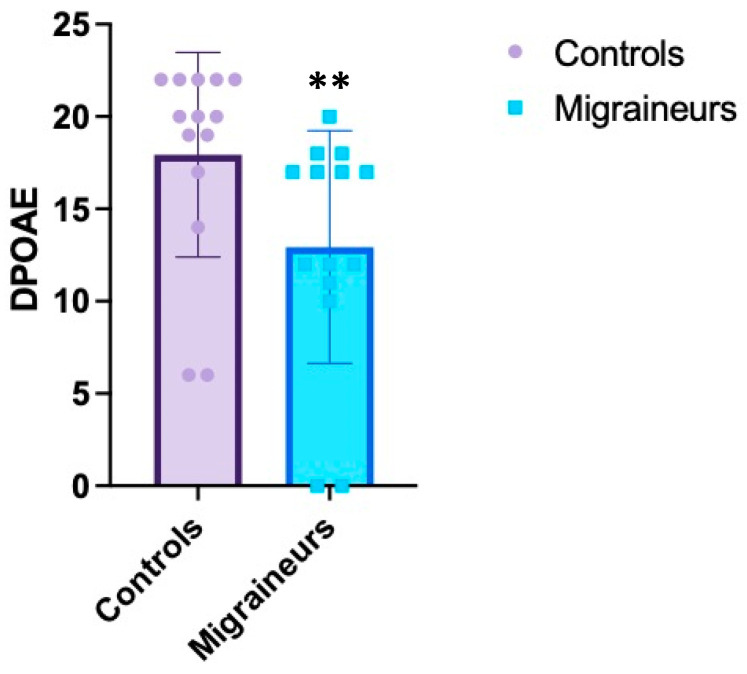
DPOAE Control versus migraine. The graph shows the differences in the value of DPOAE’ scores (*y*-axis) between migraineurs and healthy subjects. ** indicates *p* < 0.0001.

**Figure 2 jcm-10-05007-f002:**
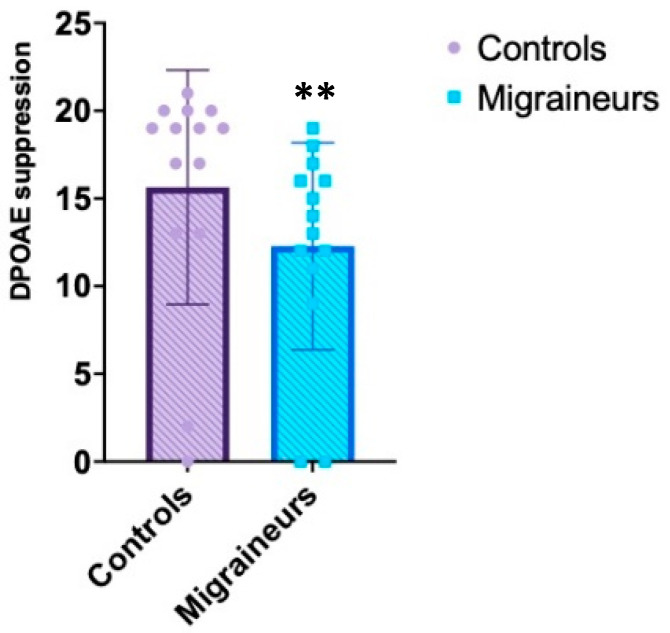
DPOAE suppression Control versus migraine. The graph illustrates the differences in the value of DPOAE suppression’ scores (*y*-axis) between migraineurs and healthy subjects. ** indicates *p* < 0.0001.

**Figure 3 jcm-10-05007-f003:**
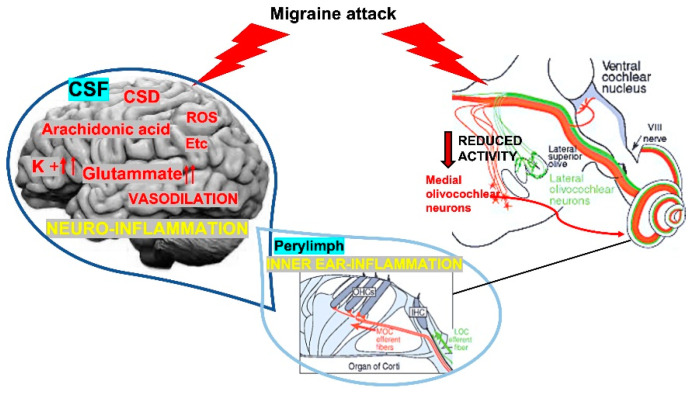
The image summarizes our hypotheses. In case of increase of pro-inflammatory cytokines, the elements migrate from brain into the inner ear through the liquid exchange between cerebrospinal fluid (CSF) and perilymph (left side of the image). Several components (Reactive Oxygen Species—ROS—, cortical spreading depression—CSD—, K+ increase, glutamate increase, arachidonic acid, vasodilation, etc.) induce an inflammation with consequent hypofunction of the inner ear cells. On the right side, the suppression of MOC activity affects the inner ear cells motility. In both cases, DPOAE will be reduced as we identified in our study.

**Table 1 jcm-10-05007-t001:** Demographic and clinical characteristics of the studied groups.

Demographic and Clinical Characteristics	Migraineurs (*n* = 50)	Controls (*n* = 58)
**Male/female**	15/35	20/38
**Age, years**	18–53 (36 ± 10)	18–53 (35 ± 8)
**Duration of illness, years**	3–40 (17 ± 12)	
**Attacks frequency (number per month)** **Low frequency (≤4/month)** **High frequency (>4/month)** **Migraine-related symptoms**	24 (48%) 26 (52%)	
**Ocular**	38 (76%)	
**Auditory**	11 (22%)	
**Dopaminergic** **Allodynia** **Use of triptans**	22 (44%) 24 (48%) 21 (42%)	

**Table 2 jcm-10-05007-t002:** OAE testing results in controls and migraineurs.

	TEOAE		TEOAE Suppression		DPOAE		DPOAE Suppression	
	Left	Right	Left	Right	Left	Right	Left	Right
Controls	15.50 ± 4.6	15.95 ± 4.56	13.8 ± 4.46	15.38 ± 4.49	13.62 ± 6.96	15.64 ± 7	12.42 ± 6.37	14.73 ± 6.98
(*n* = 58)								
Migraineurs	15.49 ± 4.53	15.24 ± 5.41	14.07 ± 4.7	14.32 ± 4.96	10.12 ± 7.88	9.33 ± 7.95	8.24 ± 8.25	8.26 ± 7.85
(*n* = 50)								
*p* value	0.990	0.462	0.756	0.249	**0.01 ***	**0.00003 ***	**0.004 ***	**0.00002 ***

Data are presented as mean ± standard deviation. * Statistically significant difference, *p* < 0.05.

**Table 3 jcm-10-05007-t003:** Correlation between audiological variables and clinical characteristics.

VARIABLES	Adjusted OR (95% CI)	*p*-Value
**DPOAE**		
Disease Duration	0.020 (0.001–0.580)	0.023
Absence of aura	0.025 (0.001–0.466)	0.013
Ocular symptoms	357.553 (2.114–60,461.505)	0.025
Auditory symptoms (ear fullness)	1100.476 (2.612–4.636)	0.023
Monthly painkiller intake (*n*)	0.007 (0.000–0.670)	0.033
**DPOAE suppression**		
Disease Duration	0.076 (0.007–0.771)	0.029
Dopaminergic manifestations	10.960 (0.980–122.529)	0.052
Allodynia	25.136 (1.710–369.467)	0.019
Not use of Triptans	0.024 (0.002–0.281)	0.003

## Data Availability

The study database is available from the Corresponding Author on reasonable request.

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
