# Peer review of "Distortion Product Otoacoustic Emissions and Their Suppression as Predictors of Peripheral Auditory Damage in Migraine: A Case-Control Study"

_jcm, 2021, doi:10.3390/jcm10215007_

Round 1

Reviewer 1 Report

This study looks at auditory efferent function in migraineurs using OAEs and suppression effects. This has been done before, but results have been mixed so more data is helpful to the field. Also the methodology and population studied is a little different to previous so there is some novel contribution.

Overall the article is adequately structured and presented, but there are a few issues with English and some language editing is needed but the manuscript is mostly comprehensible. There are some formatting issues in Table 2 and one character describing a statistics test in Figure legend 1 is a nonsense character in my script.

How were the control group identified and recruited?

The manuscript says the patients were “admitted” which sounds like this was an inpatient population, which seems unlikely. Can this be clarified?

There is an extensive literature on vestibular testing in migraine, beyond simply caloric testing, rather more than is implied in the introduction. Could this section be developed?

In terms of discussing the results of this study, there are other studies on OAE suppression in migraine which have different results. At least two have shown reduced TEOAE suppression (Joffily 2016 Murdin Laryngoscope 2010). The authors do mention one of these papers; but don’t discuss explicitly why they did not observe an effect reported in two other studies. Perhaps one explanation is that the broader cohort used here (the other studies observed subpopulations of migraineurs) is insensitive to the effect of TEOAE suppression. Is there a relevant difference between TEOAE suppression and DPOAE suppression?

Line 262…why would DPOAE be a useful screening tool? How sensitive or specific would it be? In what context could it be of any use for screening?

 Line 273 … do the authors mean “subclinical” rather than “clinical”?

Table 2 should include absolute mean values and a confidence interval,  as well as the p values.

Line 272 Is it possible to distinguish between the effects of disease duration and age in this context since age will correlate (albeit broadly) with disease duration, and DPOAE amplitude also shows some age effects?

Author Response

Thank you for the careful review of our manuscript titled, “Distortion Product Otoacoustic Emissions and their suppression as predictors of peripheral auditory damage in migraine: a case-control study”.

We are grateful for the opportunity to submit an updated manuscript, and we have revised the paper

in accordance with these valuable recommendations. We thank the reviewers for their service and

expertise, which has allowed us to materially improve the content and clarity of the submission.

We have addressed all critiques and discussed all changes in the responses below in bolded Italic. We highlighted in yellow the changes. 

Reviewer 1:

1) This study looks at auditory efferent function in migraineurs using OAEs and suppression effects. This has been done before, but results have been mixed so more data is helpful to the field. Also the methodology and population studied is a little different to previous so there is some novel contribution.  Overall the article is adequately structured and presented, but there are a few issues with English and some language editing is needed but the manuscript is mostly comprehensible.

There are some formatting issues in Table 2 and one character describing a statistics test in Figure legend 1 is a nonsense character in my script.

We agree with the reviewer; hence, a native-English speaker revised the manuscript.

Really sorry about some stylistic and formatting errors, we modified the table 2 and Figure legend 1.  

2) How were the control group identified and recruited?

The concern is right. The control group was represented by healthy volunteers, matched for age and sex, that were recruited from hospital staff, friends, and colleagues. Potential controls were excluded if they had a history of primary or secondary headaches, a history of otologic disease or clinically significant audio-vestibular symptoms, or any active medical disorder likely to interfere with testing.  Thus, we specified this concept in the material and methods section.  

3) The manuscript says the patients were “admitted” which sounds like this was an inpatient population, which seems unlikely. Can this be clarified?

Thank you for your note. Patients were recruited from the outpatient population of the Headache Centre at Tor Vergata University Hospital of Rome. Thus, we revised the sentence in the manuscript to specify this issue.    

4) There is an extensive literature on vestibular testing in migraine, beyond simply caloric testing, rather more than is implied in the introduction. Could this section be developed?

We took note of your comment, which is appreciated and agreed on, and we developed accordingly this section in the introduction.

“…Many patients with migraine are also affected by cochleovestibular symptoms (i.e. fluctuating hearing loss, aural pressure, tinnitus, hyperacusis, vertigo, motion sensitivity and imbalance), with a prevalence of auditory problems [2]. Despite this finding, a causal link between migraine and auditory discomfort remains elusive and the authors have proposed several explanations for their coexistence. Vascular, neurochemical, genetic, cortical and central sensory processing have been speculated as causes of this observation [3]. Migraineurs present prolonged interpeak latencies in the auditory brainstem evoked responses (ABR) especially when they have documented hearing loss or vertigo, which might arise from peripheral or central origin [Hamed et al, 2012]. Although patients with migraine complaining of hearing loss, pure tone audiometry and speech discrimination scores do not reveal any impairment in the most of cases [Bolay et al, 2008]. Various abnormalities have been also reported in several vestibular testing such as electronystagmography, caloric examination, autorotation test, Romberg's test, tandem Romberg test, and Fukuda stepping test; these alterations were present during the attack and in the interictal period, suggesting a possible peripheral inner ear involvement as cause of the labyrinth dysfunction [Dash et al, 2007]…”.

5) In terms of discussing the results of this study, there are other studies on OAE suppression in migraine which have different results. At least two have shown reduced TEOAE suppression (Joffily 2016 Murdin Laryngoscope 2010). The authors do mention one of these papers; but don’t discuss explicitly why they did not observe an effect reported in two other studies. Perhaps one explanation is that the broader cohort used here (the other studies observed subpopulations of migraineurs) is insensitive to the effect of TEOAE suppression. Is there a relevant difference between TEOAE suppression and DPOAE suppression?

Thank you for prompting us to clarify this point. 

There are previously published data on OAE suppression in migraineurs, but they were related only to the TEOAE examination and to a small-sized sample. Although the total suppression amplitude was lower in the individuals with migraine, these alterations were significantly observed only in a subset of migraineurs, such as women with phonophobia at the selected frequencies of 1-1.5 kHz [Joffily et al, 2016] or patients with vestibular migraine [Murdin et al, 2010]. We believe that in our study, the broader cohort might have contributed to the lack of difference in the TEOAE suppression between groups. Moreover, there are some relevant differences between TEOAEs and DPOAEs and, consequently, between their suppression magnitudes [8,9]. In fact, TEOAEs are known to originate from the activation of whole cochlea and provide information limited to middle-frequency ranges 1–4 kHz. However, cochlear alterations in the high frequencies do not significantly alter the amplitudes of TEOAEs. Compared to TEOAEs, DPOAEs precisely detect cochlear dysfunction in a frequency-specific manner and are superior in measuring larger and higher frequency ranges whereas measurement in the range of low frequencies is less reliable because of noise contamination [8,9].

TEOAEs can be detected up to a hearing loss of approximately 30 dB, but provide only little information about the frequency specificity of a possible hearing loss [8]. By contrast, DPOAEs may exist even in the case of severe hearing impairment (of up to 50 dB) and are more lasting over time being still detectable two hours post-mortem in experimental studies. Thus, both TEOAEs and DPOAEs should be used as they are complementary to each other, and both tests were employed in our study for a detailed cochlear evaluation in migraineurs [8,9].

We added the following sentences in the discussion to specify this topic:

“…Previous studies already analyzed the OAE suppression in migraineurs, but not using DPOAE -notable more sensitive to detect alteration especially in neuroinflammatory diseases [26]; moreover the previous studies have been conducted on small-sized sample. Although the total suppression amplitude was lower in the individuals with migraine, these alterations were significantly observed only in a subset of migraineurs, such as women with phonophobia at the selected frequencies of 1-1.5 kHz [7] or patients with vestibular migraine [27]. We think that our broader cohort might have contributed to the lack of difference in the TEOAE suppression among groups. Furthermore, there are some relevant differences between TEOAE and DPOAE and, consequently, between their suppression magnitudes [8,9]. In fact, TEOAE originate from the activation of whole cochlea and provide information limited to middle-frequency ranges 1–4 kHz. Compared to TEOAE, DPOAE allow the assessment of cochlear status in a frequency-specific manner and are very sensitive to measure larger and higher frequency ranges [8,9]. DPOAE may exist even in the case of severe hearing impairment (of up to 50 dB) and long-lasting over the time being still detectable two hours after death [8]. Because the two investigations are complementary, we used both TEOAE and DPOAE suppressions, for a detailed cochlear evaluation in migraineurs…”.

6) Line 262…why would DPOAE be a useful screening tool? How sensitive or specific would it be? In what context could it be of any use for screening?

As mentioned above, OAEs permit sensitive assessment of cochlear function and objectively monitor dynamic minute changes in cochlear responsiveness, that are undetectable by other audiological methods, before functional and significant hearing loss occurs from any cause. In fact, the OHCs are responsible for the cochlear sound amplification and, on the other hand, are capable of moving spontaneously in response to sound stimuli generating TEOAEs and DPOAEs (this movement is also known as electromotility). OAE amplitudes indicate the summed activity of OHCs. Minimal amounts of cochlear damage may cause measurable changes in OAE responses; and as scattered OHC loss accumulates, OAE amplitude decreases before significant long-term changes in pure tone thresholds. This is particularly relevant in the setting of patients with migraine and other neurological disorders, who may complain of recurrent hearing disturbances despite relative preservation of inner ear function, likely due to subtle inflammation-triggered damage. Our study showed that migraineurs classified as having normal hearing (as provided by the reported normal PTA), are at risk of cochlear and peripheral auditory impairment evidenced by lowering of DPOAE. Thus, alterations in the DPOAE recording may be a potential useful screening tool because it may be the earliest indicator of impending auditory malfunction in migraine, before the establishment of more permanent long-term hearing damage.   

We specify this concept in the text: “…Overall, these results highlight that DPOAE might be used to identify potential reversible hearing impairment in migraineurs; the presence of these alterations suggests that subtle sensorial affection may be an early sign of headache, a sort of additional “aura” detectable by electrophysiologic study only. The causes of these electrophysiological disturbances might be varying (figure 3); however, both DPOAE and their suppression test seem to be sufficiently sensitive to detect them…”.

7)Line 273 … do the authors mean “subclinical” rather than “clinical”?

We mean both clinical and subclinical picture/manifestation of migraine. We specify this concept in the text.

8) Table 2 should include absolute mean values and a confidence interval, as well as the p values.

We adjusted the table 2, as proposed. 

9) Line 272 Is it possible to distinguish between the effects of disease duration and age in this context since age will correlate (albeit broadly) with disease duration, and DPOAE amplitude also shows some age effects?

It is well known that hearing thresholds are poorer in adult men than in adult women and that thresholds worsen with age in both genders. Similarly, DPOAE amplitudes have been found to be larger in the young and larger in women than in men [Cilento et al , 2000]. In our study, there was no significant demonstrable correlation/relationship between altered DPOAEs and their suppression and either age or gender, also after normative correction. This may be due to the relatively young sample of migraineurs and their homogeneity. Nevertheless, we can speculate that migraine may aggravate age-related hearing loss and increase hearing impairment caused by daily exposure to noise over time, according to our findings that hearing dysfunction correlated with disease duration.

So, we specify this issue in the text: “…Since DPOAE amplitudes worsen with age similarly to hearing thresholds, we can also speculate that migraine may aggravate age-related hearing loss and increase hearing impairment caused by daily exposure to noise over time, according to our findings that hearing dysfunction correlated with disease duration…”.

Reviewer 2 Report

This study investigates the peripheral auditory pathway in migraineurs using otoacoustic emissions (OAEs), to detect alteration of cochlear functioning and possible relationship with disease severity. Sixty-two migraineurs and sixty matched controls were enrolled in the study and underwent a routine neuro-otolaryngology examination; self-administered questionnaires were used to evaluate self-perception of hearing disability. DPOAE and their suppression were found lower in migraineurs compared to controls and significantly related to the disease duration. This potential subclinical cochlear impairment in migraine detected by OAEs may represent the earliest sign of sensorineural damage in these patients. 

Please find my concerns below;

It would be better if authors could compare newly started migraine patients with this group to demonstrate this effect is directly related to the disease duration or severity. Another group would be triptan treated migraine patients as they discuss that triptan usage may decrease inflammatory response and the effect on the auditory nerve. If the authors can not include these groups to the study then it would be good to include this discussion as a limitation of the study.

Another point is the inflammation theory of migraine as a reason for the peripheral auditory damage. It is really interesting and there are data related to the release of proinflammatory cytokines to the csf after cortical spreading depolarisation (CSD) which is the elctrophysiological corralete of migraine aura (Science 2013, Mar 1;339(6123):1092-5. doi: 10.1126/science.1231897). It may strenghten the authors' possible explanation.

Figure 3, ROS is not the only source of inflammatory marker release in migraine (if authors means ROS as reactive oxygen species). It is better to include other sources of neuroinflammation including CSD, K+ increase, glutamate increase, Arachidonic acid, vasodilation etc to the figure 3 the figure is 

Minor; please check Results part '.....fifty-eight control subjects (mean age 35 ± 8 years; 20 males and 39 females)'. Is it 39 or 38 females?

Author Response

Thank you for the careful review of our manuscript titled, “Distortion Product Otoacoustic Emissions and their suppression as predictors of peripheral auditory damage in migraine: a case-control study”.

We are grateful for the opportunity to submit an updated manuscript, and we have revised the paper

in accordance with these valuable recommendations. We thank the reviewers for their service and

expertise, which has allowed us to materially improve the content and clarity of the submission.

We have addressed all critiques and discussed all changes in the responses below in bolded Italic. We highlighted in yellow the changes. 

Reviewer 2

1) This study investigates the peripheral auditory pathway in migraineurs using otoacoustic emissions (OAEs), to detect alteration of cochlear functioning and possible relationship with disease severity. Sixty-two migraineurs and sixty matched controls were enrolled in the study and underwent a routine neuro-otolaryngology examination; self-administered questionnaires were used to evaluate self-perception of hearing disability. DPOAE and their suppression were found lower in migraineurs compared to controls and significantly related to the disease duration. This potential subclinical cochlear impairment in migraine detected by OAEs may represent the earliest sign of sensorineural damage in these patients. Please find my concerns below.

It would be better if authors could compare newly started migraine patients with this group to demonstrate this effect is directly related to the disease duration or severity. Another group would be triptan treated migraine patients as they discuss that triptan usage may decrease inflammatory response and the effect on the auditory nerve. If the authors cannot include these groups to the study then it would be good to include this discussion as a limitation of the study.

Reviewer’s consideration is remarkable and appealing. Unfortunately, we cannot include these groups of patients, so we added it as a limitation of the study.

2) Another point is the inflammation theory of migraine as a reason for the peripheral auditory damage. It is really interesting and there are data related to the release of proinflammatory cytokines to the csf after cortical spreading depolarisation (CSD) which is the elctrophysiological corralete of migraine aura (Science 2013, Mar 1;339(6123):1092-5. doi: 10.1126/science.1231897). It may strenghten the authors' possible explanation.

According to the reviewer, we properly pointed out this matter in the manuscript and we have added the reference in the manuscript.

3) Figure 3, ROS is not the only source of inflammatory marker release in migraine (if authors means ROS as reactive oxygen species). It is better to include other sources of neuroinflammation including CSD, K+ increase, glutamate increase, Arachidonic acid, vasodilation etc to the figure 3 the figure .

According to the reviewer’s suggestion we amended the figure 3

4) please check Results part '.....fifty-eight control subjects (mean age 35 ± 8 years; 20 males and 39 females)'. Is it 39 or 38 females?

We revised the results part and clarified the point ‘…fifty-eight control subjects (mean age 35 ± 8 years; 20 males and 39 females)'. It is 38 females.
